# Continuous and Impact Cutting in Mechanized Sugarcane Harvest: Quality, Losses and Impurities

**João Vitor Paulo Testa [1], Murilo Battistuzzi Martins [2,*], Aldir Carpes Marques Filho [3], Kléber Pereira Lanças [1], Renato Lustosa Sobrinho [4], Taciane Finatto [4], Mohammad K. Okla [5] and Hamada AbdElgawad [6]**

[1]  Rural Engineering and Agricultural Mechanization Department, College of Agricultural Sciences, São Paulo State University—UNESP, Av. Universitária, 3780, Botucatu 18610-034, SP, Brazil; jvtesta@gmail.com (J.V.P.T.); planca@gmail.com (K.P.L.)

[2]  Cassilândia University Unit, Agronomy Department, Mato Grosso do Sul State University—UEMS, 306 Road, Km 6, Cassilândia 79540-000, MS, Brazil

[3]  Agricultural Engineering Department, Federal University of Lavras, Lavras 37200-000, MG, Brazil; aldircm@gmail.com

[4]  Department of Agronomy, Federal University of Technology—Paraná (UTFPR), Pato Branco 85503-390, PR, Brazil; lustosa.renato@gmail.com (R.L.S.); tfinato@gmail.com (T.F.)

[5]  Botany and Microbiology Department, College of Science, King Saud University, Riyadh 11451, Saudi Arabia; malokla@ksu.edu.sa

[6]  Integrated Molecular Plant Physiology Research, Department of Biology, University of Antwerp, 2020 Antwerp, Belgium; hamada.abdelgawad@uantwerpen.be

*  Correspondence: murbam@gmail.com

**Abstract:** Sugarcane harvesting requires improvements, particularly in cutting tools. Continuous cutting saws have been introduced as a solution to this issue. This study evaluates the performance of two basal sugarcane cutting systems in different fields: a traditional impact cut system (ICS) with knives and a continuous cut system (CCS) with saw blades. Tests were conducted during two crop cycles in three areas, using a $3 \times 2$ factorial design with two cutting devices and four replications per treatment. Cut quality indices and ratoon damage were analyzed using descriptive statistics. Raw material losses were subjected to the Shapiro–Wilk normality test, ANOVA, and Tukey's test at 5% probability. Significant differences in cutting quality were found across different areas. The total crop productivity influenced sugarcane cut quality, with the CCS showing (0.8 Mg ha$^{-1}$) visible losses in higher productivity areas, which is a 74% increase compared to the ICS. In lower productivity areas, the CCS demonstrated better loss performance (0.8 Mg ha$^{-1}$). Additionally, the stumps damage rate for the CCS was lower than that for the ICS (0.15 and 0.28, respectively), indicating that saws can preserve cane fields and enhance longevity.

**Keywords:** *Sacharum* spp.; mechanization; basecut; harvesting; cut device

## 1. Introduction

Mechanized sugarcane harvesting brought several environmental benefits, reducing the need for labor and fires [1] and increasing agricultural performance [2]. However, mechanized harvesting also generates problems, such as increased raw impurities, field losses [3], lower cut quality [4–7] and ratoons with damage [8], resulting in fragmented stumps [9].

The base cutter is a vital sugarcane harvester component. Its work efficiency was directly related to production cost per unit area and its decisive impact on sugarcane yield in the following year [10–12]. During mechanized harvesting, the stalk's sugarcane base is cut by two rotating disks equipped with cutting knives. The discs rotate along the harvest line with a convergent movement, simultaneously cutting and feeding stalks to the harvester's internal systems [13].

The cutting mechanisms wear out and lose quality according to the use intensity. The abrasive wear of the blades is influenced by several factors such as the geometry, chemical composition and hardness of the workpiece material, friction between the bodies, contact with plants, climatic and operating conditions, working speed and the size and the nature of soil particles [14].

The sugarcane quality cut determines productivity in the following year [15]. The basal cutting mechanism is a significant loss responsible for mechanized sugarcane harvest, incorporating soil mineral impurity and raw material into machines. Cutting blade wear increases damage and loss rates, which can be aggravated as the blade cutting angle decreases [16].

Many researchers developed studies with new cutting mechanisms and tools to find improvements in basal cutting processes. Voltarelli et al. [17] evaluated damage caused by coated and uncoated knives; they concluded that harvesting operation is better when using coated knives, which is explained by coated knives being less susceptible to wear [18]. Worn blades can damage cane stalks. Mello and Harris [19] pioneered the study of continuous sugarcane cutting using saws with satisfactory results. Under controlled conditions, Marques Filho et al. [13] obtained promising results for applying saws to the sugarcane basal cut. So, depending on the field condition, the saws can cause less damage in sugarcane stalks and cause less raw material loss.

This study aimed to evaluate two basal sugarcane cutting systems in different field conditions: a traditional impact cut system (ICS-knives) and a continuous cut system (CCS-saws) adapted to harvest machines. The cutting devices were evaluated regarding cutting quality, industrialized material losses, vegetal and mineral impurities to standard methodology.

## 2. Material and Methods

### 2.1. Site of Study

During two crop production cycles, the experimental research was conducted in São Paulo state, Brazil, in three locations belonging to Borborema and Lençóis Paulista municipalities. The areas may have some different characteristics; however, the average temperature and rainfall during the harvests over the last 30 years are similar [20–22]. For this study, we made an agreement with the productive properties to collect data in real field situations. In this sense, Casler (2015) in his study titled "Fundamentals of Experimental Design: Guidelines for Designing Successful Experiments" wrote the following: "Even field locations that may seem uniform, are likely not uniform, thus the correct patterns of spatial variability cannot be fully predicted without years of trial and error".

The specific characteristics of each experimental area are described in Table 1.

**Table 1.** Characteristics of experimental areas to evaluate sugarcane basal cutting mechanisms.

| Data | Area 1 | Area 2 | Area 3 |
|---|---|---|---|
| Local | Borborema—SP | Lençóis P.—SP | Lençóis P.—SP |
| Crop Variety | SP813250 | RB855156 | RB966928 |
| Harvest Date | November/2016 | April/2017 | April/2017 |
| Soil Texture | Sandy | Clayey | Clayey |
| Implantation/cut | 2010—6° Cut | 2016—1° Cut | 2014—3° Cut |
| Productivity Mg ha$^{-1}$ | 61.4 | 114.8 | 93.4 |
| Spacing | S. Simple | S. Simple | Alternating Double |
| Range (m): | 1.5 | 1.5 | 1.5–0.9 |
| Geographic Coord. | 21°37′10.41″ S 49°00′17.92″ O | 22°35′26.77″ S 48°43′56.06″ O | 22°37′02.96″ S 48°44′01.73″ O |

According to data from the Brazilian National Supply Company (CONAB) [23], in Brazil, there are planting areas classified as low, medium, and high productivity.

These classifications primarily vary based on resources and available technologies; where properties with more resources tend to use denser spacings to offset the high investments in fertilizers.

We chose properties that could represent each of the productive system profiles classified by CONAB (low, medium, and high productivity). Our methodological objective was to encompass the main production systems used for the crop in Brazil, aiming to produce data that reflect the real conditions of commercial cultivation of the sugarcane crop in the country.

Two basal cutting systems were tested under field conditions during the sugarcane harvest: the continuous cutting system (CCS), composed of circular saw blades, and the impact cutting system (ICS) composed of carbon steel knives (Figure 1).

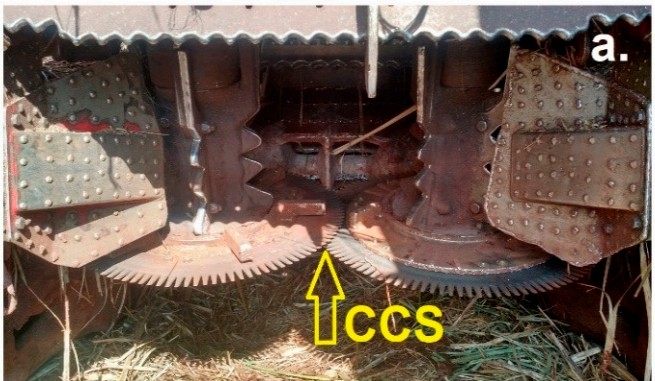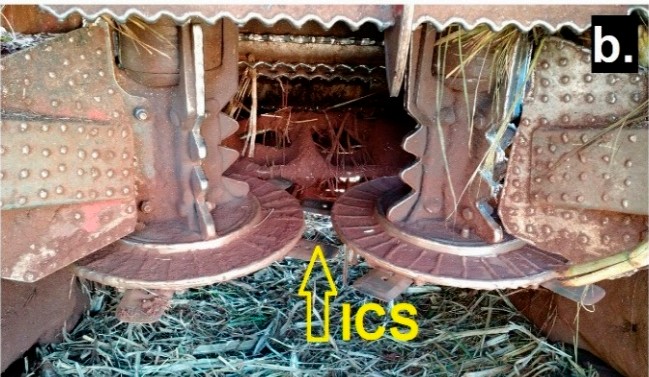

**Figure 1.** Sugarcane cutting systems. (**a**) Continuous cutting system (CCS), composed of steel saws; (**b**) Conventional impact cutting system (ICS), consisting of carbon steel cutting knives.

The continuous cutting system (CCS) consisted of segmented saws of the "Kruger" brand, model KG3NES, mounted on two trays specific to the manufacturer, each tray having six sectors of semicircular saws, each sector having 21 cutting teeth with a thickness of 0.004 m (Figure 1a). In the impact cutting system (ICS), steel knives of the "Unimil" brand, 6-hole model, with sharpening on four sides and 0.00475 m thick, were used. The complete cutting device had 2 circular trays and 5 mismatched knives in each tray (Figure 1b).

### 2.2. Machines Models Harvesting

Two sugarcane harvesters' models were used. In area 1, a John Deere harvester, model 3520, manufactured in 2015, with 3.749 engine hours and 2.232 hs of the elevator, was used. In Areas 2 and 3, the harvester used was a CASE IH, model A8800 SR, manufactured in 2015, with 3.555 engine hours and 1.840 elevator hours.

We are aware that different locations could produce "subtle" differences in lignification and stem diameter, which could potentially influence the cutting efficiency. Precisely with this consideration in mind, we chose to standardize the forward speed at an average common to the three productive scenarios under study. This decision results from the observation that excessively high forward speeds may decrease the cutting efficiency, while moderate to low speeds tend to increase both the effectiveness and power of the cut.

Thus, all machines in this study were set to work at an average speed of 4.5 km h$^{-1}$ by maintaining a single forward speed; we also sought to mitigate the possible effects of any local variation on the cutting effectiveness. The base cutting mechanism had its angular speed set to 480 rpm on each rotating tray, regardless of cutting tool (knives or saws). This configuration was adopted because it is the recurring average operating condition in sugarcane harvesting operations.

*2.3. Visible Losses of Raw Material*

The visible losses were quantified in a 10 m$^2$ area after the harvesting machine passage, covering two crop rows. For the conventional spacing (1.5 m between rows), an area of 3 × 3 was used, and 3 m for the alternate double spacing, 2.4 m wide and 4.15 m long, was used to demarcate the sample area with four replications.

To assess raw material loss level to sugarcane productivity, the losses in Mg ha$^{-1}$, the methodology proposed by Benedini, Brod and Perticarrari [24] was followed, where losses smaller than 2.5% are considered at the "low" level, between 2.5 and 4.5% are considered "medium" and above 4.5% are considered "high." Raw material losses in Mg ha$^{-1}$ were converted to percentages according to Equation (1):

$$Pd = \frac{Pc}{P + Pc} \times 100 \tag{1}$$

where:

$Pd$ represents losses (%);
$Pc$ represents visible raw material losses in the field (Mg ha$^{-1}$);
$p$ is sugarcane productivity (Mg ha$^{-1}$).

*2.4. Basal Cut Quality*

The ratoons were evaluated after the harvester's passage to determine basal cut quality. Ten sugarcane stumps from each sample plot were randomly evaluated, and the damage index was determined by firm, medium or plucked ratoon stalks. The damage index was obtained according to the methodology of Ripoli et al. [25]. To evaluate ratoon damage, the human evaluator, with his hands, applied a force to the remaining stalks, inferring movement resistance and giving a score according to plant element motricity (Table 2).

**Table 2.** Ratoon damage classification.

| Test Score | Weight | Characteristics |
|:---:|:---:|:---:|
| 3 | 0.0 | Firm stalks with little movement |
| 2 | 0.5 | Stalks with medium movement |
| 1 | 1.0 | Torn or loose stalks |

These evaluations, as they are based on subjective criteria, depended on the evaluator's expertise; therefore, to minimize personal human effects in the variability results obtained, all evaluations were performed by the same researcher.

Each evaluator score applied was interpolated with its respective weight through Equation (2), where the values were provided to ratoon damage index ($i_A$).

$$i_A = \frac{p_{SF} \times n_{SF} + p_{SM} \times n_{SM} + p_{SA} \times n_{SA}}{n} \tag{2}$$

where $i_A$ represents the ratoon damage index;

$p_{SF}$ represents the weight attributed to firm ratoons or with little movement;
$n_{SF}$ represents the number of firm ratoons or ratoons with little movement;
$p_{SM}$ represents the weight attributed to firm ratoons with medium movement;
$n_{SM}$ represents the number of firm ratoons with average movement;
$p_{SA}$ is the weight attributed to the ratoons pulled out or released;
$n_{SA}$ represents the number of ratoons pulled out or released;
$n$ represents the total ratoons number examined.

*2.5. Quality of the Harvested Raw Material*

Samples were taken directly on the elevator exit harvester to evaluate raw material quality; this material passed through all cleaning machine systems, thus allowing evaluating the amount of foreign material (strange matter vegetal and mineral). After removal, the

sample was separated into fractions to determine plant impurity; pointers fractions, leaves, straw and roots were separated.

We classified mineral and plant impurity indices following the methodology proposed by Benedini, Brod and Perticarrari [24], and the reference values are described in Table 3.

**Table 3.** Mineral and vegetal impurities classification.

| Classification | Mineral Impurity | Vegetal Impurity |
| --- | --- | --- |
| Low | <0.4% | <4% |
| middle | 0.4% a 0.6% | 4% a 6% |
| High | >0.6% | >6% |

Adapted from Benedini, Brod and Perticarrari (2013) [24].

*2.6. Experimental Design and Statistical Result Analysis*

The experimental design used was a $3 \times 2$ factorial scheme comprising three experimental areas (areas 1, 2 and 3) and two cutting devices, continuous and impact (saws and knives), with four replications in each treatment. Cut quality indices and ratoon damage were subjected to descriptive statistical results analysis based on applied methodology. The average losses of industrialized material were subjected to the Shapiro–Wilk normality test, analysis of variance (ANOVA) and Tukey test at 5% probability; all statistical analyses were performed in Minitab® software version 16.

## 3. Results and Discussion

Significant differences are found in industrialized losses material to cutting systems and field areas (Figure 2). Except in area 1, the continuous cutting system (CCS) shows the highest amounts of visible losses, the highest value in area 2 (8 Mg ha$^{-1}$) and the lowest value in area 1 (0.8 Mg ha$^{-1}$). We are able to observe that the CCS system performs better in areas with lower productivity, as the amount of material to be processed is reduced.

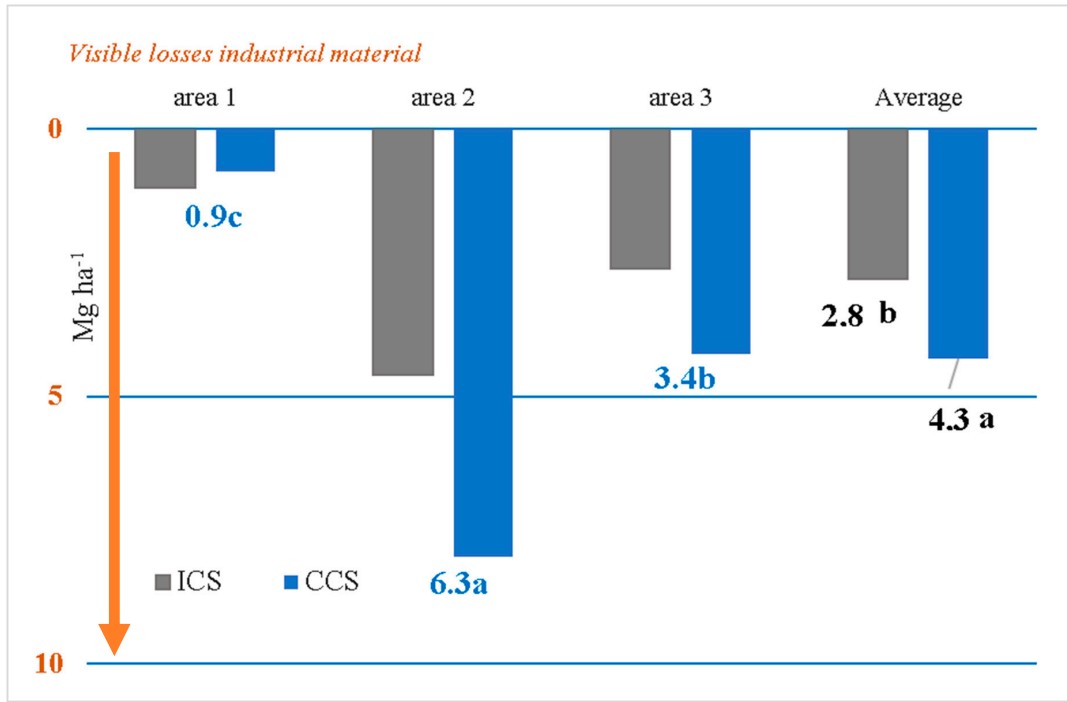

**Figure 2.** Total losses averages in the field areas for each cutting device. ANOVA: F test (area): 30.75 *; F test (cutoff): 7.73 *; F test (interaction): 3.44 ns; CV (%): 30.7. Means followed by the same letters (lowercase blue box—loss/area) and average uppercase tool do not differ by Tukey's test ($\alpha = 5\%$), CV: variation coefficient, * significant at 5% probability and ns not significant at 5% probability.

We found that in the most productive sugarcane fields, the performance of the continuous cutting system led to more significant losses; this may have occurred due to the time the cutting mechanism takes to work on the stumps, which is longer than the ICS impact system. Passion et al. [3] evaluated different sugarcane basal cutting systems, and they concluded the cut process presents instability regarding quality and damage levels, depending on the tool used.

In the ICS, visible industrialized material losses were smaller than in the CCS in areas 2 and 3. In area 2, there was the most significant difference in proportion between the systems tested, and the CCS showed losses 74% greater than the ICS. This finding indicates that the saws had operating limitations in high-productivity areas, because with continuous cutting, the entire surface of the discs is in contact with the stalks and causes longer cutting times, which can generate an unwanted new cut of the stalk that is not collected by the machine and increases losses.

Isolating the area factor, we have a difference between the basal cutting mechanisms. We verified that the losses went from 2.8 to 4.3 Mg ha$^{-1}$, increasing 1.5 Mg ha$^{-1}$ of visible losses. These visible losses are mainly related to the inefficiency of the continuous cutting process in the high-productivity areas, where the saws drag the stalks and bring the entire cane to the ground.

According to Benedini, Brod and Perticarrari [24], the relative losses of the ICS were considered low in area 1 and medium in areas 2 and 3. For the CCS, there were medium relative losses in area 3 and high losses in area 2. Manhães et al. [26], evaluating the sugarcane harvest quality of a JD3520 harvester with a basal impact cutting system (ICS), found losses of 12, 7 and 6 Mg ha$^{-1}$ at different operating speeds, which were values higher than those obtained in this work for the ICS.

In area 1, the cutting systems showed similar total losses and relative loss levels; however, in areas 2 and 3, the loss levels increased for both systems; these areas had more productive sugarcane fields, which made the cutting and harvesting process difficult, and increasing losses. As they had more productivity, it is possible that there was a greater amount of stalks per clump, imposing a greater amount of material to be cut, and this material was not used for processing inside the machine and resulted in increased losses.

Momin et al. [27], in an evaluation of different basal cutting tools in sugarcane, with a harvester working at 6.3 km.h-1, found results more significant than 83% of culms without damage, 11.3% partially damaged and 5.65% significantly damaged in the continuous cutting system with a serrated tool. Bernache et al. [16] found an increase in the damage rate caused by the wear of cutting tools but did not find a relationship with crop regrowth.

In the CCS, the increase that occurred was superior to that of the ICS, demonstrating a greater susceptibility of this basal cutting tool to the difficulties imposed by areas 2 and 3. Paixão et al. [8], evaluating the quality harvesting process in impact cutting systems, inferred that machine model and operators training could interfere with process quality. Corredo et al. [28] stated that improvements in electronic systems and real-time sensors are needed to control loss conditions better.

Silva et al. [29], studying ICS cutting systems in commercial sugarcane fields, found damage rates similar to those in our research. The authors evaluated the harvest in burnt and unburned sugarcane fields (green) and indicated that minor damage occurs to raw harvested sugarcane fields; this is fundamentally due to the reduced harvest speed in sugarcane fields with a greater plant size. Kumar et al. [2] showed several benefits in cover soil maintenance, indicating that it interferes with crop longevity. In sugarcane, burning improves machine performance but causes serious environmental problems.

Marques Filho et al. [13] showed that various raw material losses depend on the cutting tool. Invisible losses can reach 0.52 and 0.54% of the total harvested. The authors indicated more significant invisible losses for the impact-cutting system with knives.

When relatively compared, we classified the visible raw material losses to standardized ranges (Figure 3). Saws, unlike knives, cannot break the surface soil layer or perform

planting line sweeping, leaving sugarcane fractions inside the furrow and greater length stalks, resulting in a significant increase in visible losses.

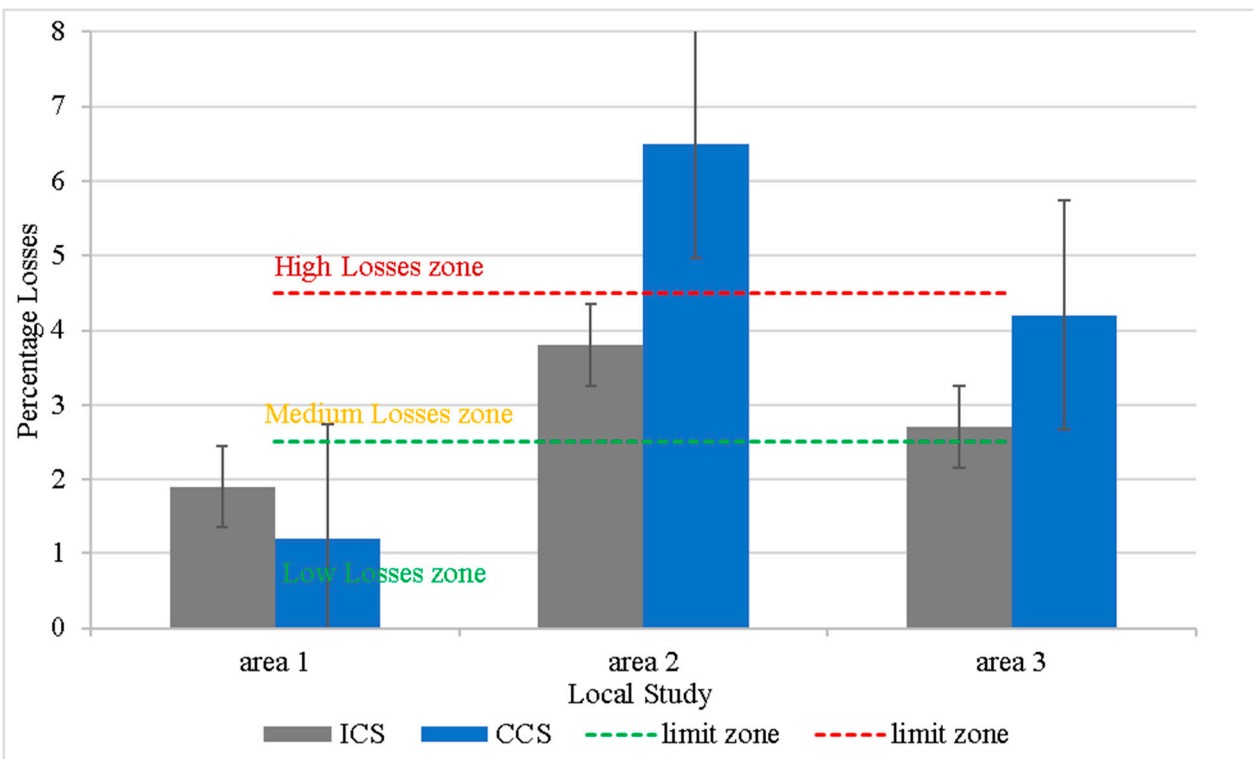

**Figure 3.** Visible material losses to total productivity; limit zones of material loss according to Benedini, Brod and Perticarrari (2013) [24].

In areas 2 and 3, the losses percentage caused by the CCS reached critical levels compared to the ICS. In area 1, with a lower total productivity of 61.4 Mg ha$^{-1}$, the cutting systems are not overloaded and losses are significantly lower, in addition to the fact that with greater productivity, the culms may not be all erect and aligned, making the process of cutting and guiding the material into the machine more difficult. Manhães et al. [26] found percentage losses between 8 and 16% in a cane field with average productivity of 74.7 Mg ha$^{-1}$, indicating that cane field size contributes to increased raw material losses.

In this study, we established an operational speed pattern (4.5 km h$^{-1}$) for the harvesting standards production unit regardless of the device cutting. While it is not considered an ideal speed, it is a usual speed among Brazilian producers in the sugarcane harvesting process; however, changes may occur depending on the conditions available at the time of harvest.

However, increasing the speed can affect the cutting tools' performance. Martins and Ruiz [11], evaluating raw material losses in sugarcane crops, found no differences for this parameter as a function of operational speed. Marques Filho et al. [13] showed that harvest speed could increase raw material losses. In this study, harvester speeds did not vary significantly. However, our results show high losses for continuous cutting mechanisms in areas of higher productivity. These losses could be accentuated at speeds higher than that those adopted in this study.

Ding et al. [30], studying cutting height sensors, indicated that the base cutting depth error was directly related to the advanced machine speed. The authors found better harvesting conditions with a 1 km h$^{-1}$ speed; these were lower than the averages in this study and barely applicable to field practices. Low speeds impair operational performance and machines designed to harvest satisfactorily well above 3 km h$^{-1}$. Wang et al. [6] showed that cut quality is a severe problem in sugarcane seed production. In this way, they

applied machine learning to improve operation; this application can also improve the base cut harvesting operation, opening perspectives for future investigations.

The indices of damage to the stumps, calculated according to a specific methodology, can be seen in Figure 4. In all experimental areas, knives presented a higher damage index to stumps. On average, for area treatments, there was a damage index of 0.50 for knives and saws and an index of 0.27. The lower damage rates indicate that crop regrowth is probably favored in cane fields with continuous cutting because there are no fissures that can attack pests, diseases and healthier development, making it possible, in some applications, to achieve a greater longevity of the cane field.

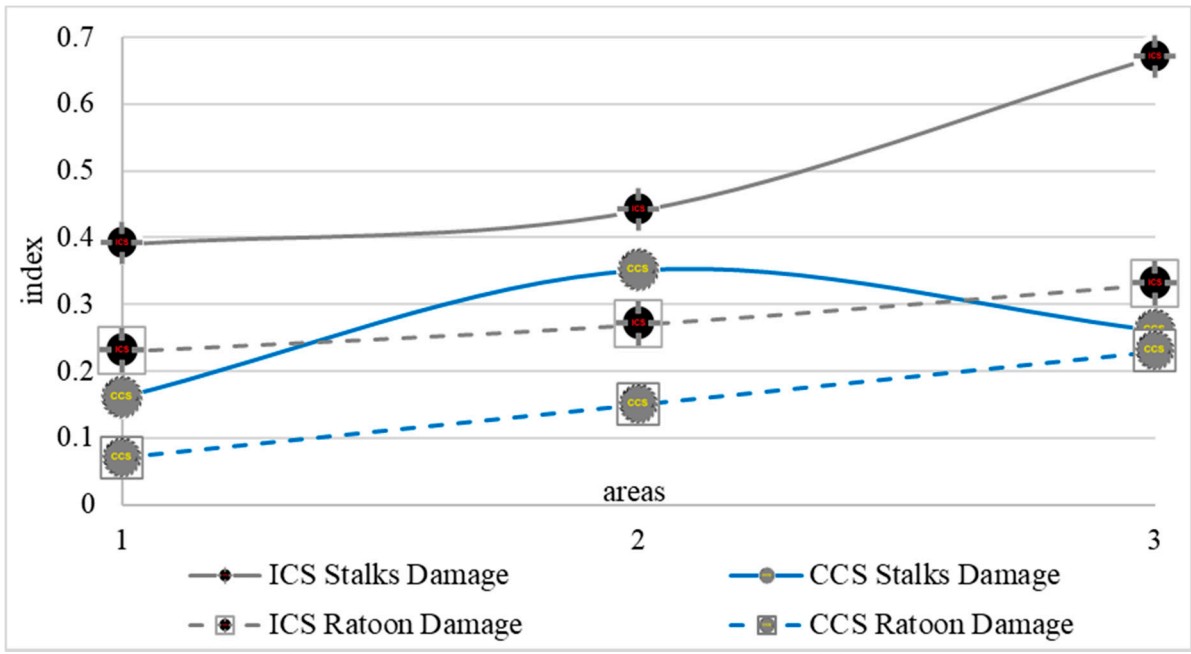

**Figure 4.** Stalks and ratoons damage index.

Momin et al. [27] stated that the choice of inefficient and low-quality cutting systems could compromise the sugarcane field's longevity and reduce the crop's potential. Inadequate cuts cause fractures in the stumps remaining in the soil, preventing future sprouting; therefore, cutting with knives becomes risky in high-productivity sugarcane systems. Qiu et al. [4] indicated an optimal combination between the harvester's forward speed and the cutting tool's rotation so that high speeds tend to cause splits in the culms, accelerating the stump's deterioration.

In area 2, a high damage rate is observed in the saws test (0.40). We noted the slightest difference between the evaluated systems (0.04); this more significant amount of damage presented by the saws was due to the arrangement of clumps in the valley, which made harvesting more difficult for the saws, similar to industrialized raw material.

In area 3, there was the point with the greatest significant distance between the systems (0.41) and the highest damage index observed for the knives during the test (0.67), showing that the knives had difficulty cutting in the double alternating spacing, indicating a greater difficulty in aligning the knives and the crop line. Wang et al. [7] showed that some machine devices might be responsible for increased crop losses. Thus, the angle of the cane and the lifting mechanism are fundamental in the shock rates of stumps.

According to Marques Filho et al. [13], in the CCS, the cutting time is extended to the ICS system; this causes the culm to flex just before cutting, causing damage and generating more chips and sawdust. Liu et al. [31] and Srivastava et al. [32] explain this phenomenon based on the principle of viscoelastic behavior of plant material, where the cutting tool pressure initially causes a plant stem deformation. This deformation is a function of the

contact time and the cutting thickness tool, causing failures in vegetable fibers, generating their complete rupture.

In addition, crop characteristics affect harvest quality. In a single row, machines push plants in adjacent rows, falling in the opposite return direction machine path due to the rank's divider. Thus, a sugarcane crop can be classified as favorable to harvest initially, but after the harvester passes, this condition can be changed by the action of row dividers [16].

Damage ratoons had a similar tendency to cut quality tests; knives presented a higher damage index in all tested areas; however, the difference in the tested systems' cut occurred more significantly for tested sites. Three tested areas show significant differences, except for area 1 (Figure 4). The average ratoons damage index values for saws and knives were 0.15 and 0.28, respectively.

Yang et al. [33] found a high correlation between mathematical simulation models and sugarcane base shear stresses. The authors stated that the shear stresses propagate underground and reach the roots at moderate intensity. In this way, more significant cutting impacts, as obtained in the ICS, can affect root growth and increase raw material losses.

There was no significant difference between the cutting systems tested for the averages of foreign plant matter. The differences were significant only between the tested areas (Figure 5); this is because the cutting system has less influence on the foreign plant material, with the cleaning systems having the greatest impact on this variable. However, there is a relationship with the cutting system, since that starts the process of sending the material into the machine, thus being an indispensable variable in the study. In absolute terms, the saws (CCS) provided more plant foreign matter in the most productive sugarcane fields, areas 2 and 3, compared to the ICS.

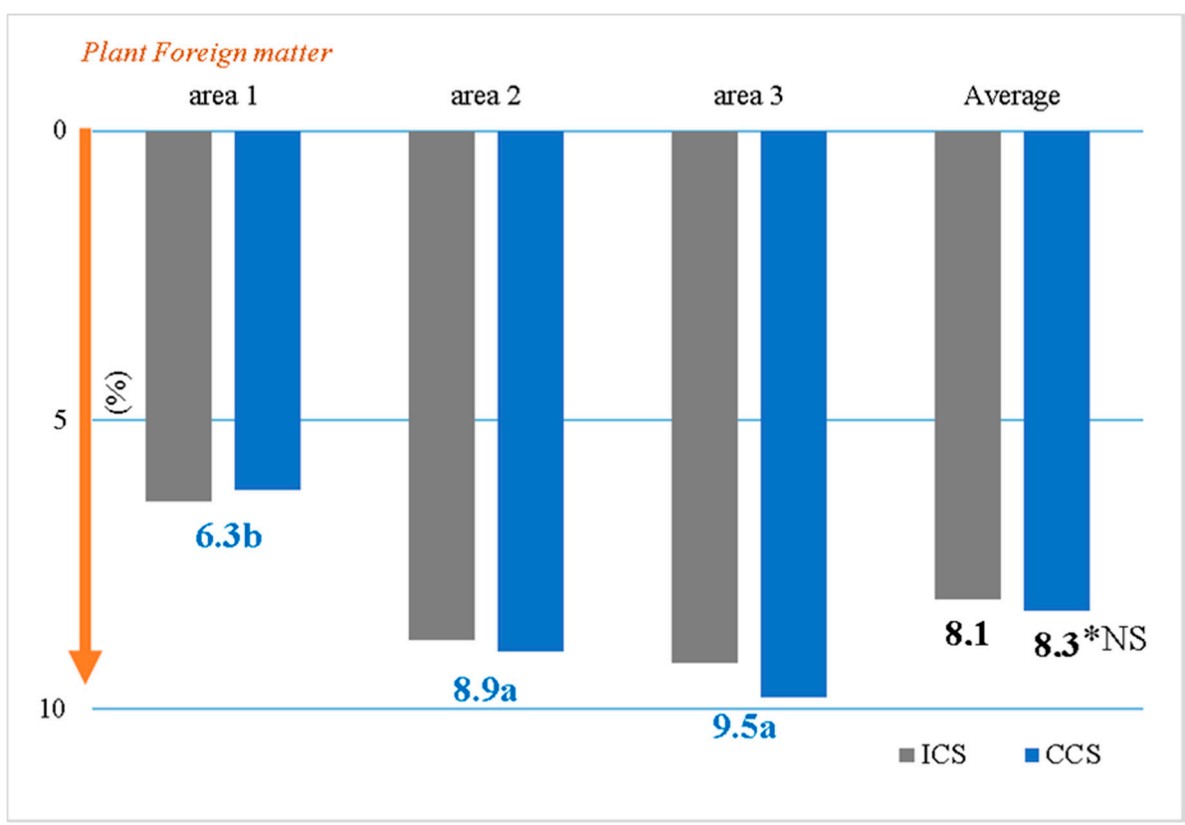

**Figure 5.** Plant foreign matter (%) in the harvested material. ANOVA: means by the same letters do not differ by Tukey's test ($\alpha$ = 5%), CV—Deviation Coefic. 10.2%, * significant at 5% probability and NS not significant at 5% probability. F test (area) 31.5 *; F test (s.cut) 0.31 NS; F test (interaction) 0.37 NS.

The lowest foreign plant matter occurred in area 1, with an average of 6.3%, and the highest amount of foreign plant matter occurred in areas 2 and 3, which is expected, since the determination of plant matter is relative to the productivity of the area. According to the classification proposed by Benedini, Brod and Perticarrari [24], the level of foreign plant matter was considered high in all treatments, as it was above 6%.

There was no significant difference between the cutting systems tested on foreign mineral matter; the differences were significant only between the tested areas (Figure 6). According to Benedini, Brod and Perticarrari [24], the mineral impurity found in area 1 was low, while those in areas 2 and 3 was classified as medium. According to studies by Mello et al. [34], mineral impurities cause industry losses, leading to metallic elements corrosion and incrustations in boilers.

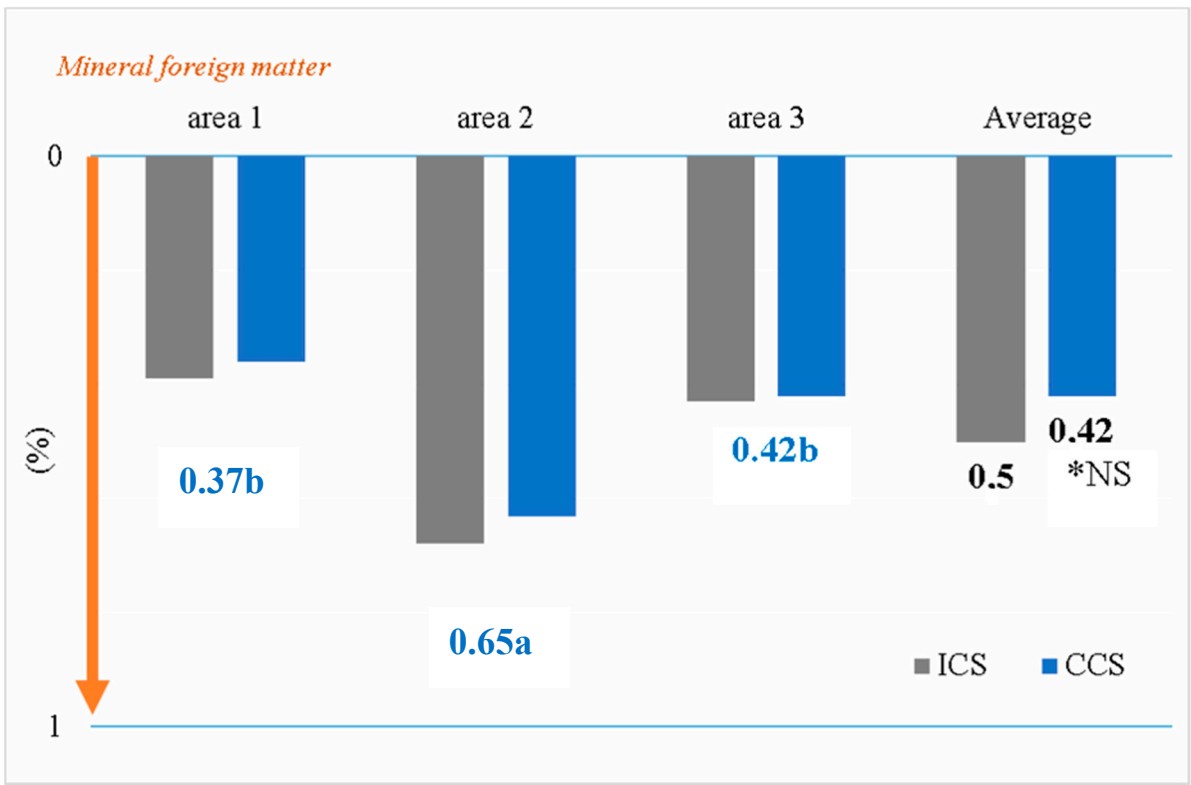

**Figure 6.** Averages of mineral foreign matter present in the raw material. ANOVA: Means by equal lowercase letters do not differ by Tukey's test ($\alpha = 5\%$), CV—Deviation Coefic. 15.1%, * significant at 5% probability and NS not significant at 5% probability. F test (area) 30.23 *; F test (s.cut) 0.97 NS; F test (interaction) 0.17 NS.

Ding et al. [30] attributed the base cutter of the harvesters to a significant impact on losses in the field. Our work showed that in addition to the cutting system, the tool model directly affects losses under different operating conditions since the ICS can more easily break the surface layer of the soil to carry out the cut and, consequently, drag a greater amount of mineral material along with the stalks. Qiu et al. [4], studying sugarcane cutting quality factors, found that the most important factors were the cutting tool angle, speed harvester, and cutting base rotation. In our research, the rotation remained at the standard harvester without variation; however, manufacturers can increase the machines with models that allow an alteration of this parameter.

During the harvesting process, factors such as the topography and slope of the area and adjustments in the mechanical systems of the harvester can lead to an increase in impurities in the raw material [34]. In this way, the correct systematization and leveling of the area are necessary to reduce mineral impurities. Additionally, modern height control

mechanisms are being developed to provide less damage during cutting, which can help reduce future crop losses and increase process efficiency [30].

Li et al. [5], in research using a virtual prototype of sugarcane cutting, revealed that the base's rotation speed and the machine's advanced speed affect the cutting time. Our raw material loss results are linked to the cutting time of each evaluated mechanism.

Evaluative studies related to the losses in each harvesting system are fundamental for increasing the productive activity and improving the processes. Corredo et al. [28], in an extensive review of sensors and monitoring systems in machines, stated that the application of real-time sensors to determine losses during the harvesting process, still in the machine, is fundamental for reducing losses and increasing efficiency, as the real-time data collection allows adjustments in the harvesting and cleaning mechanisms.

Our study aimed to provide an overview of the performance of two cutting systems selected for each reality of productive systems classified by CONAB-BRASIL [23]. We believe that future studies can better explore the effects of different advancement speeds, and we are considering this for our next research.

## 4. Conclusions

We rate two sugarcane-cutting systems: the traditional impact cut and the new continuous system. We detailed the quality cut and the losses under field conditions, opening new perspectives for investigations on cutting tools in sugarcane and applying saws to sugarcane cutting machines' performances.

Significant differences were found in cutting quality depending on the type of sugarcane crop. The crop's total productivity interfered in the sugarcane cut's quality. In the higher productivity areas, the CCS system showed more visible losses (8 Mg ha$^{-1}$), representing an increase of 74% in the ICS and indicating that in very dense cane fields, the CCS continuous cutting mechanism is inefficient in cutting and collecting plant material.

Nonetheless, in lower productivity areas, the CCS improved its performance to the ICS (0.8 Mg ha$^{-1}$). In this way, the more significant productivity areas suffer if cut with saws. The stumps damage rate by the CCS was lower than that of the ICS at 0.15 and 0.28, respectively, indicating that saws can preserve cane fields and increase longevity.

**Author Contributions:** A.C.M.F. and J.V.P.T. conceived the research. M.B.M. and K.P.L., conducted experiments. R.L.S., T.F., M.K.O. and H.A. contributed material. J.V.P.T., M.K.O. and H.A. analyzed the data and conducted statistical analyses, while final revisions of the work were made by R.L.S. and M.B.M. All authors have read and agreed to the published version of the manuscript.

**Funding:** This research was funded by the Researchers Support Project number (RSP-2023/374) of King Saud University, Riyadh, Saudi Arabia.

**Institutional Review Board Statement:** Not applicable.

**Data Availability Statement:** The datasets used and/or analyzed during the current study are available from the corresponding author on reasonable request.

**Acknowledgments:** The authors wish to thank the College of Agricultural Sciences, São Paulo State University—UNESP, State University of Mato Grosso do Sul—BraziL. The authors expend their appreciation to the Researchers Support Project number (RSP-2023/374) of King Saud University, Riyadh, Saudi Arabia.

**Conflicts of Interest:** The authors declare no conflict of interest.

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
