# Peer review of "Continuous and Impact Cutting in Mechanized Sugarcane Harvest: Quality, Losses and Impurities"

_agriculture, doi:10.3390/agriculture13071329_

Round 1

Reviewer 1 Report

The detailed geometry and dimensions of the two cutting system, especially the size of the toothed blade, should be given in the ms. 

Author Response

The devices were better described in their constructive forms and manufacturers. We have improved the methodology for better understanding.

Reviewer 2 Report

This paper addresses the effect of different cutting tools on the cutting effect and tool wear during the harvesting of sugarcane. This study evaluates the performance of two basal sugarcane cutting systems in different fields: a traditional impact cut system (ICS) with knives and a continuous cut system (CCS) with saw blades. The results obtained were that in the areas of higher productivity, the CCS system showed more visible losses. And in lower productivity areas, CCS showed better loss performance. The manuscript can provide a reliable basis for the design of sugarcane harvesting machinery.

(1) The format of the title in the manuscript is not standardized. Suggest reworking.

(2) The main novelty and contributions of this work should be better explained.

(3) Each section of the manuscript is short and unfulfilling, resulting in overall logical confusion throughout the text.

(4) The meaning of the letters of the formula in the manuscript is not explained properly. It is suggested to revise.

(5) Figures 2 and 6 in the manuscript are not drawn in a standard way, resulting in confusing image content. It is recommended to check and revise the whole text.

Minor editing of English language required

Author Response

We standardized the title and changed the format for clarity and conciseness. We highlight the novelties of this work in the Introduction. We changed the letters of the formula for standardization and modified the figures. The ms has been reformatted as per the guidelines for improvement.

Reviewer 3 Report

It is well know that the cutting tools play very important roles in the sugarcane harvesting process. This study evaluates the performance of two basal sugarcane cutting systems in different fields: a traditional impact cut system (ICS) with knives and a continuous cut system (CCS) with saw blades. Significant differences in cutting quality were found across different areas. Research has shown that the areas of greater productivity suffer if cut with saws. The lower damage index indicates that saws can preserve cane fields and increase longevity.

However, there are some aspects throughout the manuscript needed to be improved:

First, I don't think that the title of this manuscript can cover the contents and logical relationship of the full text. So, the best way, I think, is to revise the title to enhance generality of this manuscript.

Second, in the "Results and Discussion" section, there were so many analysis from other researchers. Correspondingly, the author's own views are few and the analysis is not deep enough. Meanwhile, theoretical analysis is superficial, which leads to the fact that the innovation of this manuscript is not outstanding.

Third, there's one detail that makes me wonder: this manuscript mention that there is an optimal combination between the harvester's forward speed and the cutting tool's rotation. why the speed is chosen as 4.5 km h-1 and the angular speed is 480 rpm in the study, correspondingly? Does this mean that the two parameters are optimal according to the research results? What happens if other parameters are used?

The last one, in the "conclusions" section, I found that it is too brief to reflect the innovation and integrity of this study. Further expansion and enrichment for the conclusions are very necessary and important.

The quality of English should be improved and perfected.

Author Response

All suggested changes were considered in the manuscript.

Reviewer 4 Report

- In this research, the 3x2 factorail design was used for field trials. It was indictaed that first factor is 3 different sugar can areas and the other is  2 different cuttings mechanizms. It is think that although the usage of the cutting mechanizms is suitable for a factor, the usage of the 3 different areas shown as a factor is not.  Because the experiments were carried out in different years including first is 2016, the others 2017 years, and these areas has many differents characteristics. It can be said that the main factor effecting the comparison parameters used in the study is the feeding of the machines. This factor emerges as field yield/productivity due to using of only one forwarding speed in this study. The figures expalined this opinion. 

Three different speed values could be used for this purpose in only one area for the other factor instead of three area. Or, the experiments could be conducted in three different fields had same properties and had different yield for a more healthy comparison.

- The figure should have been prepared more careful.

Author Response

We improved the quality of the Introduction and better justified the paper's research gap;

The determination of the factorial occurs by evaluating the performance of 2 cutting tools as a function of 3 different profiles of sugarcane crops. It is precisely because of the differences between the areas that we consider an independent factor. In this way, we were able to highlight the performance of the tools in different field conditions; this enriches the quality of the research.

We modified the figures and their standardization to improve the presentation of results.
Please see the attachment.

Round 2

Reviewer 3 Report

I think the author has made more in-depth modification and improvement for this manuscript, and it has met the requirements for publication. It was my suggestion to publish it after a slight revision.

The quality of English language is minor editing of English language required.

Author Response

Dear Reviewer,

Thank you for your feedback on the manuscript. We appreciate your evaluation and have made several modifications and improvements to the article based on it.

Once again, thank you for your valuable contributions.

Best regards,

Renato

Reviewer 4 Report

Dear Editor,

If it is appropriate for you, I think it would be more appropriate for me not to reevaluate the article, as I have declared the decision as reject. I think that it seems more appropriate to make a decision by considering the other reviewers' opinions.

With my best regards.

Author Response

We appreciate your thorough review and valuable comments.

In regard to the 3x2 factorial design used in the study, we understand your concerns about the representativeness of different sugarcane cultivation sites as a factor in the experiment.

In this sense, we agree with your suggestions about the importance of considering the uniformity of the experimental fields.

However, it must be emphasized that for this study we made an agreement with the productive properties in order to collect data in real field situations, and this brought several complexities, thus, it was not possible to conduct the study in the three areas within the same year.

We are aware that the areas may have some different characteristics, however, the average temperature and rainfall during the harvests over the last 30 years are similar.

Source: https://www.climatempo.com.br/climatologia/2236/borborema-sp

Source: https://www.climatempo.com.br/climatologia/477/lencoispaulista-sp

Furthermore, according to data from the Brazilian National Supply Company (CONAB, 2023), in Brazil there are planting areas classified as low, medium, and high productivity.

These classifications primarily vary based on resources and available technologies; where properties with more resources tend to use denser spacings to offset the high investments in fertilizers.

Therefore, we chose properties that could represent each of the productive system profiles classified by CONAB (low, medium, and high productivity).

Our methodological objective was to encompass the main production systems used for the crop in Brazil, aiming to produce data that reflect the real conditions of commercial cultivation of the sugarcane crop in the country.

In this sense, Casler (2015) in his study titled "Fundamentals of Experimental Design: Guidelines for Designing Successful Experiments" wrote the following:

"Even field locations that may seem uniform, are likely not uniform, thus the correct patterns of spatial variability cannot be fully predicted without years of trial and error".

We are aware that different locations could produce "subtle" differences in lignification and stem diameter, which could potentially influence the cutting efficiency. Precisely with this consideration in mind, we chose to standardize the forward speed at an average common to the three productive scenarios under study. This decision results from the observation that excessively high forward speeds may decrease the cutting efficiency, while moderate to low speeds tend to increase both the effectiveness and power of the cut.

Thus, by maintaining a single forward speed, we also sought to mitigate the possible effects of any local variation on the cutting effectiveness.

Therefore, our study could provide a general overview, evaluating the two chosen cutting systems in each of these realities of productive systems.

We believe that future studies can better explore the effect of different forward speeds, and we are considering this for our next research.

As mentioned earlier, the aim of this specific study was to get a general idea of the behavior of these two cutting systems in the different productive realities classified by CONAB-BRAZIL.

We appreciate your comments on the figures and all of them have been altered after the first review in response to your feedback, to ensure that they conveyed the information in a more precise and effective way.

We hope that these explanations and the changes we have made in our manuscript align with your suggestions. We are open to any other comments that may help us further improve our work.

Best regards,

Renato

REFERENCES

Casler, M.D. (2015), Fundamentals of Experimental Design: Guidelines for Designing Successful Experiments. Agronomy Journal, 107: 692-705. https://doi.org/10.2134/agronj2013.0114

Acompanhamento de Safra Brasileira de Cana-de-Açúcar; Companhia Nacional de Abastecimento—(CONAB): Brasília, Brazil, 2023.
